# Transcriptome change of *Staphylococcus aureus* in infected mouse liver

Hiroshi Hamamoto [1,8], Suresh Panthee [2,3,8], Atmika Paudel[4], Suguru Ohgi[5,6], Yutaka Suzuki[7], Koichi Makimura[1] & Kazuhisa Sekimizu [2✉]

We performed in vivo RNA-sequencing analysis of *Staphylococcus aureus* in infected mouse liver using the 2-step cell-crush method. We compared the transcriptome of *S. aureus* at 6, 24, and 48 h post-infection (h.p.i) in mice and in culture medium. Genes related to anaerobic respiration were highly upregulated at 24 and 48 h.p.i. The gene expression patterns of virulence factors differed depending on the type of toxin. For example, hemolysins, but not leukotoxins and serine proteases, were highly upregulated at 6 h.p.i. Gene expression of metal transporters, such as iron transporters, gradually increased at 24 and 48 h.p.i. We also analyzed the transcriptome of mouse liver infected with *S. aureus*. Hypoxia response genes were upregulated at 24 and 48 h.p.i., and immune response genes were upregulated from 6 h.p.i. These findings suggest that gene expression of *S. aureus* in the host changes in response to changes in the host environment, such as the oxygenation status or immune system attacks during infection.

[1] Teikyo University Institute of Medical Mycology, 359 Otsuka, Hachio-ji shi, Tokyo 192-0395, Japan. [2] Drug Discoveries by Silkworm Models, Faculty of Pharma-Science, Teikyo University, 359 Otsuka, Hachio-ji shi, Tokyo 192-0395, Japan. [3] GenEndeavor LLC, 26219 Eden Landing Rd, Hayward, CA 94545, USA. [4] International Institute for Zoonosis Control, Hokkaido University, North 20, West 10, Kita-ku, Sapporo Hokkaido 001-0020, Japan. [5] Laboratory of Microbiology, Graduate School of Pharmaceutical Sciences, The University of Tokyo, 7-3-1 Hongo, Bunkyo-ku, Tokyo 111-0033, Japan. [6] Kyowa Kirin Co., Ltd., 1-9-2 Otemachi, Chiyoda-ku, Tokyo 100-0004, Japan. [7] Department of Computational Biology and Medical Sciences, Graduate School of Frontier Sciences, The University of Tokyo, 5-1-5 Kashiwanoha, Kashiwa shi, Chiba 277-8562, Japan. [8]These authors contributed equally: Hiroshi Hamamoto, Suresh Panthee. ✉email: sekimizu@main.teikyo-u.ac.jp

The rapid emergence of multi-antimicrobial resistant strains has created an urgent need for the development of novel therapeutic agents. The number of antibacterial agents with novel mechanisms of action recently approved by the FDA is limited[1], however, suggesting the need for new strategies to discover target molecules that can be developed as novel antimicrobials[2]. Antimicrobials are typically screened on the basis of their antimicrobial activity in vitro. As recently pointed out[3, 4], however, pathogen behavior exhibited during in vitro culture is much different from that in vivo in the host; thus, it is important to identify antimicrobial targets expected to be more efficient in the host.

*Staphylococcus aureus* has successfully adapted to the environmental conditions of the human body. It can survive and respond to various conditions in the human body, and causes a wide range of diseases, such as acne, pneumonia, and bacteremia[5], despite its relatively small genome size[6]. In addition, *S. aureus* easily acquires multidrug resistance[7]. The estimated number of deaths caused annually by infection with methicillin-resistant *S. aureus* is 10,000 in the United States and 4000 in Japan[8]. *S. aureus* secretes various kinds of toxins, such as hemolysins, leukotoxins, and proteases[9], and grows under both aerobic and anaerobic conditions[10]. Furthermore, *S. aureus* has at least 5 iron acquisition systems and many metal transporters that are essential for its colonization and pathogenesis under host conditions[11,12]. Many genes have been identified as pathogenic factors of *S. aureus*, and analysis of their transcriptomes under host infection conditions is crucial. By performing RNA sequencing (RNA-Seq) analysis, we revealed the in vivo transcriptome of *S. aureus* Newman strain at various time-points in the liver of mice systemically infected with *S. aureus*, in contrast to previous studies in which analysis was performed at only one specific time-point after systemic infection, or at acute and chronic time-points in local infection. For example, the transcriptional profile of *S. aureus* in the kidneys was examined at 48 h post-infection (h.p.i.) following intravenous inoculation in mice with decreased susceptibility[13]; acutely in a local prosthetic joint infection[14]; and at acute (7 days) and chronic (28 days) time-points in murine osteomyelitis[15]. In the present study, we selected a highly virulent strain that killed the animals within 72 h.p.i., and evaluated the transcriptome at 6 h.p.i., when mice exhibited no symptoms, and at 24 and 48 h.p.i., when mice exhibited a phenotype of reduced locomotion and some individuals begin to die. We recently established an improved in vivo RNA-Seq analysis applicable to a smaller *S. aureus* population size in infected organs by taking advantage of the fact that Gram-positive bacteria can be separated from host cells by mechanical disruption due to the presence of a strong cell wall. This technique has been successfully applied to in vivo RNA-Seq for *Streptococcus pyogenes*[16], and in the present study, we applied the method to *S. aureus* in a systemic mouse infection model.

## Results and discussion

### In vivo RNA-Seq analysis for *S. aureus* infection in mouse liver.

We previously reported that the 2-step cell-crush method was applicable to in vivo RNA-Seq analysis of *S. pyogenes*, which have a rigid cell wall like *S. aureus*, in necrotizing fasciitis[16]. In this method, the first step is to use large beads to crush and lyse the host tissue in lysis buffer, followed by the use of small beads to crush enriched *S. aureus* cells in mouse organs to prepare enriched bacterial RNA for RNA-Seq analysis (Fig. 1). In the present study, we performed in vivo RNA-Seq analysis of *S. aureus* grown in organs of mice systemically infected with *S. aureus*. Injection of *S. aureus* Newman strain into the mouse tail vein killed half of the mice within 48 h.p.i., and all the mice within 72 h.p.i. (Fig. 2a). Under this condition, the number of bacterial cells per organ increased exponentially in the kidney and heart within 24 h.p.i., and reached $10^7$ colony-forming units (CFU)/mg in the liver at 6 h.p.i., with more than $10^6$ CFU/mg in most individuals remaining at 48 h.p.i. (Fig. 2b). The number of bacteria in the blood, on the other hand, was not high at any time during the experiment. Thus, we performed in vivo RNA-Seq analysis of *S. aureus* in the liver at 6, 24, and 48 h.p.i., and obtained approximately 160 to 700 thousand uniquely mapped reads on the *S. aureus* genome (Supplementary Table 1). The number of genes with no mapped reads was 97 (3% of all genes) for RNA extracted from in vitro culture and 374 (12% of all genes) in *S. aureus* Newman strain isolated from liver 6 h.p.i. Therefore, most of the genes in the *S. aureus* genome were successfully analyzed by this method (Supplementary Data 1). We found 472, 545, and 616 genes with significant upregulation (>2-fold) and 637, 558, and 655 genes with significant downregulation (>2-fold) at 6, 24, and 48 h.p.i., respectively.

### Pathway analysis of the genes altered after infection with *S. aureus*.

To elucidate the trend of *S. aureus* gene expression in the host environment, we performed a KEGG pathway enrichment analysis to compare the in vivo gene expression with that in culture medium in vitro (Table 1). The expression of genes involved in carbon metabolism (glycolysis) and TCA cycle pathways was significantly upregulated at 6 h.p.i., but not at 48 h.p.i (Table 1). Expression of genes involved in beta-oxidation (responsible for the production of acetyl-CoA from fatty acids), and the PTS system (required for incorporation of phosphorylated saccharide) was significantly upregulated starting at 24 h.p.i. Expression of genes required for iron acquisition, such as the biosynthesis of various secondary metabolites—staphyloferrin A and B—was not upregulated at 6 h.p.i., but was upregulated after 24 h.p.i. In addition, expression of ABC transporters (required for the acquisition of nickel and manganese) was upregulated after 24 h.p.i. On the other hand, expression of genes involved in terpenoid backbone biosynthesis (required for cell wall synthesis, pigment, menaquinone, and peptidoglycan biosynthesis) was downregulated from 24 h.p.i. As for the host side, we performed RNA-Seq analysis using RNA of liver organs mixed in a sample

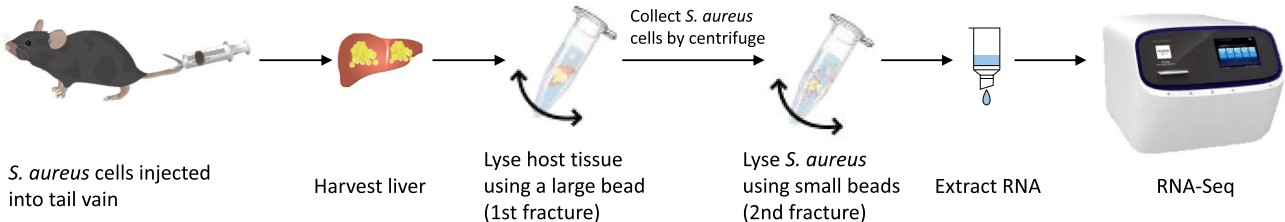

**Fig. 1 The 2-step cell-crush method for in vivo RNA-Seq analysis of *S. aureus* in mouse liver.** C57BL/6J mice were infected with *S. aureus* Newman strain injected through the tail vein, and organs were harvested at 6, 24, and 48 h.p.i. Tissue and bacterial cells were separately crushed and lysed in lysis buffer using beads of 2 different sizes.

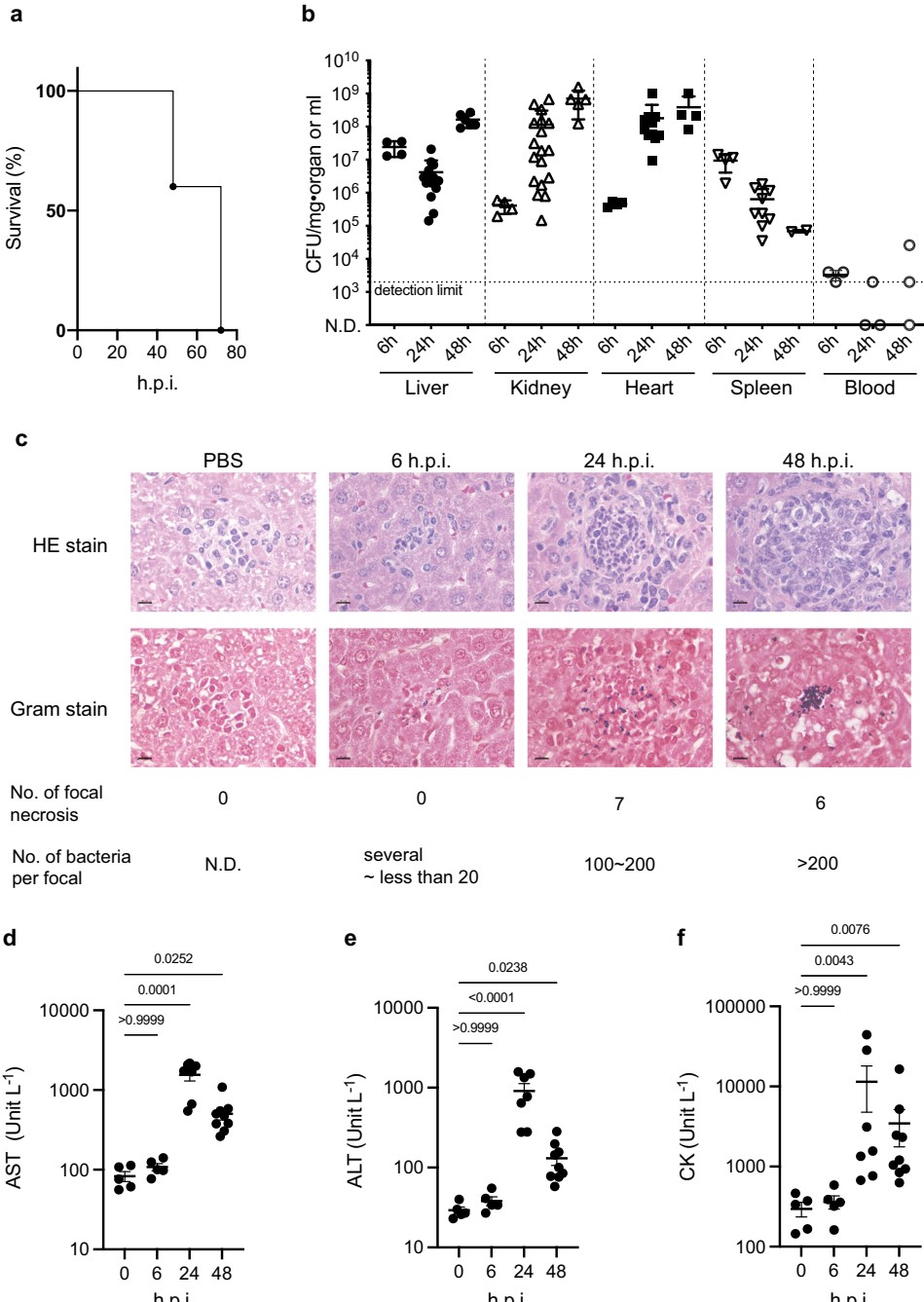

**Fig. 2 Conditions of systemic *S. aureus* infection model in this analysis. a** Survival of mice infected with *S. aureus* Newman strain. Bacterial cells (5.6 × 10^7 CFU) were injected into the tail vein (*n* = 5). Data represent the mean ± SEM. **b** Number of *S. aureus* cells in each organ after infection with (5.3 × 10^7 CFU, *n* = 2 to 11). **c** Histopathological analysis of liver from mice infected with *S. aureus* Newman strain (6.8 × 10^7 CFU). Photographs are representative of the results for 2 mice. The bars in the pictures were indicated as 10 μm. **d**–**f** Analysis of hematological parameters of infected mice. AST (**d**), ALT (**e**) and CK (**f**) parameters in the plasma collected from mice (*n* = 5 for 0 and 6 h.p.i., *n* = 7 for 24 h.p.i. and *n* = 9 for 48 h.p.i.) infected with *S. aureus* Newman strain (6.8 × 10^7 CFU) were measured. Data represent the mean ± SEM and statistical analysis was performed using a Kruskal-Wallis test with Dunn's multiple comparisons test.

that mapped uniquely on the mouse genome (Supplementary Data 2) and selected genes with a significant 5-fold difference (false discovery rate [FDR] *p*-value < 0.05) in liver infected with *S. aureus* compared with liver injected with phosphate-buffered saline (PBS) to perform the GO term enrichment analysis (Reactome, Supplementary Data 3). The results suggested that genes involved in the induction of innate immunity and inflammation, and those related to metal sequestration were

significantly upregulated (Supplementary Table 2). On the other hand, reactomes related to ATP synthesis, such as respiratory electron transport, ATP synthesis by chemiosmotic coupling, and heat production by uncoupling proteins, were downregulated at all time-points (Supplementary Table 2). In addition, *hif1a* and *hif3a*, which are involved in the hypoxia response, were upregulated after 24 h.p.i., suggesting that the host environment was low in oxygen at the late stage of infection. To further confirm the

**Table 1 KEGG pathway enrichment analysis for upregulated and downregulated genes of S. aureus-infected liver compared with culture medium.**

| Hours | Pathway ID | Description | GeneRatio | BgRatio | FDR p-value |
|---|---|---|---|---|---|
| *Upregulated* | | | | | |
| 6 h | sae00020 | Citrate cycle (TCA cycle) | 15/157 | 22/963 | 0.0000 |
| | sae01120 | Microbial metabolism in diverse environments | 43/157 | 132/963 | 0.0000 |
| | sae01200 | Carbon metabolism | 29/157 | 80/963 | 0.0001 |
| 24 h | sae00997 | Biosynthesis of various secondary metabolites—part 3 | 9/208 | 10/963 | 0.0004 |
| | sae01120 | Microbial metabolism in diverse environments | 48/208 | 132/963 | 0.0006 |
| | sae00290 | Valine, leucine, and isoleucine biosynthesis | 9/208 | 11/963 | 0.0006 |
| | sae00020 | Citrate cycle (TCA cycle) | 13/208 | 22/963 | 0.0018 |
| | sae01210 | 2-Oxocarboxylic acid metabolism | 11/208 | 18/963 | 0.0033 |
| | sae00650 | Butanoate metabolism | 10/208 | 16/963 | 0.0041 |
| | sae05150 | Staphylococcus aureus infection | 17/208 | 38/963 | 0.0077 |
| | sae02060 | Phosphotransferase system (PTS) | 12/208 | 23/963 | 0.0077 |
| | sae00071 | Fatty acid degradation | 7/208 | 10/963 | 0.0085 |
| | sae01100 | Metabolic pathways | 125/208 | 497/963 | 0.0204 |
| | sae02010 | ABC transporters | 28/208 | 85/963 | 0.0396 |
| 48 h | sae00220 | Arginine biosynthesis | 15/253 | 18/963 | 0.0000 |
| | sae02010 | ABC transporters | 39/253 | 85/963 | 0.0008 |
| | sae00997 | Biosynthesis of various secondary metabolites—part 3 | 9/253 | 10/963 | 0.0008 |
| | sae00340 | Histidine metabolism | 12/253 | 16/963 | 0.0008 |
| | sae00290 | Valine, leucine, and isoleucine biosynthesis | 9/253 | 11/963 | 0.0020 |
| | sae01230 | Biosynthesis of amino acids | 42/253 | 100/963 | 0.0021 |
| | sae01210 | 2-Oxocarboxylic acid metabolism | 12/253 | 18/963 | 0.0027 |
| | sae02060 | Phosphotransferase system (PTS) | 14/253 | 23/963 | 0.0030 |
| | sae02024 | Quorum sensing | 28/253 | 61/963 | 0.0032 |
| | sae05150 | Staphylococcus aureus infection | 19/253 | 38/963 | 0.0068 |
| | sae00052 | Galactose metabolism | 12/253 | 20/963 | 0.0069 |
| | sae00071 | Fatty acid degradation | 7/253 | 10/963 | 0.0216 |
| | sae01100 | Metabolic pathways | 148/253 | 497/963 | 0.0284 |
| | sae02020 | Two-component system | 29/253 | 75/963 | 0.0398 |
| *Downregulated* | | | | | |
| 6 h | sae03010 | Ribosome | 47/297 | 73/963 | 0.0000 |
| | sae00230 | Purine metabolism | 25/297 | 43/963 | 0.0041 |
| 24 h | sae03010 | Ribosome | 46/311 | 73/963 | 0.0000 |
| | sae00240 | Pyrimidine metabolism | 18/311 | 28/963 | 0.0125 |
| | sae00900 | Terpenoid backbone biosynthesis | 9/311 | 11/963 | 0.0195 |
| 48 h | sae03010 | Ribosome | 56/460 | 73/963 | 0.0000 |
| | sae00900 | Terpenoid backbone biosynthesis | 11/460 | 11/963 | 0.0065 |
| | sae00550 | Peptidoglycan biosynthesis | 20/460 | 24/963 | 0.0065 |
| | sae00240 | Pyrimidine metabolism | 22/460 | 28/963 | 0.0115 |

host condition, we performed a histopathological analysis and biochemical tests to evaluate liver function. We observed a number of abscesses at 24 and 48 h.p.i., but none at 6 h.p.i. In addition, the number of bacteria per abscess increased in a time-dependent manner (Fig. 2c). Furthermore, hematological parameters, such as aspartate aminotransferase activity (AST), alanine aminotransferase activity (ALT), and creatine kinase activity (CK) in the plasma, which are indicators of liver and heart damage, were significantly increased at 24 and 48 h.p.i. (Fig. 2d–f), suggesting that the liver and heart were damaged within 24 h after *S. aureus* infection. These results well correlated with the trend of the *S. aureus* gene expression changes in the host and suggested that carbon metabolism, metal metabolism, and cell wall synthesis of *S. aureus* were highly influenced by the host condition.

**Energy metabolism**. We further analyzed the gene expression changes in each *S. aureus* pathway in mouse liver after infection. Expression of genes involved in the glycolysis pathway was suppressed in mouse liver compared with the culture medium conditions (Supplementary Fig. 1) throughout the infection period. Expression of genes involved in the TCA cycle was relatively upregulated until 24 h.p.i. and downregulated at 48 h.p.i. On the

other hand, fermentation-related genes such as *pflB*, *ldh*, and *adhE* were highly upregulated at 24 and 48 h.p.i. In addition, the genes involved in nitrate respiration, such as *narK*, which is required for nitrate uptake; the *narGHJI* operon encoding respiratory nitrate reductase; and the *nirBD* operon encoding assimilatory nitrite reductase, which produces nitrate ($NO_3^-$), an electron acceptor instead of oxygen in anaerobic conditions in the electron transport chain (Fig. 3a)[17], were upregulated at 24 h.p.i. and highly upregulated at 48 h.p.i. (Fig. 3b). These results suggest that nitrate respiration was upregulated at the late stage of infection, during which the bacterial number was increasing in liver abscesses, and damage was occurring in the liver and heart (Fig. 2c–f). The increased nitrate respiration may be a response of *S. aureus* in the liver to an oxygen insufficiency associated with abscess formation and liver tissue damage. We disrupted the *narK* gene to evaluate the requirement of nitrate respiration in the virulence of *S. aureus* and found the disruptant strain reduced virulence, although, the complemented strain did not recover the virulence of the disruptant strain (Supplementary Fig. 2a). Further studies are needed to clarify the requirement of genes involved in nitrate respiration for *S. aureus* virulence. On the other hand, the mouse-killing ability of the *pflB* gene-disrupted mutant was not significantly reduced (Supplementary Fig. 2b).

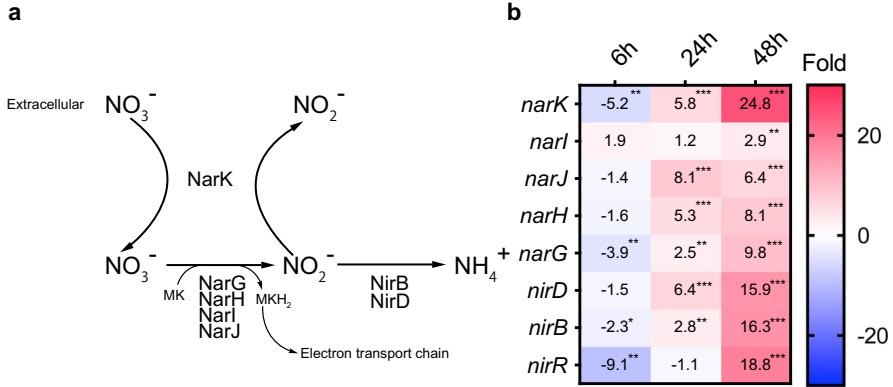

**Fig. 3 Gene expression changes regarding nitrate respiration of *S. aureus* infected in liver. a** Metabolic pathway of nitrate in *S. aureus*. **b** Expression changes in genes involved in the nitrogen respiration pathway. The values in the box show fold-change compared with *S. aureus* cultured in the TSB medium and the boxes. The asterisk *. **, and *** indicate FDR *p*-values < 0.05, <0.01, and <0.001, respectively.

**Table 2 Gene expression change of *S. aureus* lipid metabolism in the host liver compared with culture medium.**

| Gene | KEGG pathway | 6 h.p.i. | | 24 h.p.i. | | 48 h.p.i. | |
|---|---|---|---|---|---|---|---|
| | | Fold change | FDR *p*-value | Fold change | FDR *p*-value | Fold change | FDR *p*-value |
| *fadA* | Fatty acid degradation[a] | 12.0 | <0.001 | 11.5 | <0.001 | 6.7 | <0.001 |
| *fadB* | | 26.3 | <0.001 | 16.8 | <0.001 | 7.3 | <0.001 |
| *fadD* | | 84.7 | <0.001 | 61.9 | <0.001 | 9.7 | <0.001 |
| *fadE* | | 23.4 | <0.001 | 30.0 | <0.001 | 7.7 | <0.001 |
| *fab* | Fatty acid elongation[b] | −1.8 | 0.050 | −2.3 | 0.006 | −5.3 | <0.001 |
| *fabD* | | −1.9 | 0.030 | −2.3 | 0.005 | −3.7 | <0.001 |
| *fabG* | | −1.8 | 0.024 | −2.8 | <0.001 | −3.2 | <0.001 |
| *fabH* | | −5.3 | <0.001 | −4.7 | <0.001 | −4.3 | <0.001 |
| *fabI* | | −1.1 | 0.959 | −1.5 | 0.214 | −3.2 | <0.001 |
| *fabZ* | | −2.7 | <0.001 | −4.1 | <0.001 | −4.5 | <0.001 |

[a]https://www.kegg.jp/kegg-bin/show_pathway?sae00071.
[b]https://www.kegg.jp/kegg-bin/show_pathway?sae00061.

PflB is involved in synthesizing acetyl-CoA from pyruvate in anaerobic conditions, although other enzymes also cover this enzymatic reaction. Thus, an alternative pathway might compensate for its function in the *pflB* gene disruption mutant.

**Lipid metabolism.** KEGG pathway analysis suggested the upregulation of genes regulating fatty acid degradation. The genes *fadABDE* are involved in lipid beta oxidation and required for acetyl-CoA production utilized in the TCA cycle. These genes were upregulated in the host liver from the initial stage of infection compared with culture medium conditions (Table 2). In contrast, expression of the *fabIH* genes, required for type II fatty acid synthetase, a target for antimicrobial agent development[18], was significantly downregulated in the later stage of infection (Table 2). These findings suggest that *S. aureus* infected in mouse liver obtained a part of its energy from fatty acid degradation.

**Metal acquisition system.** Iron acquisition is essential for pathogen growth in the host[19]. *S. aureus* has at least 5 iron acquisition systems[20,21], and the genes involved in these systems are known to be upregulated in the host because the pathogen-infected host hides iron by increasing metal sequestration proteins, as shown in Supplementary Table 2. Although the expression of iron acquisition system genes other than *sbnCD* was not upregulated at 6 h.p.i., it was highly increased at 24 and 48 h.p.i. (Fig. 4a), a pattern that did not correspond to that of the host's metal sequestration proteins, which were upregulated from 6 h.p.i. It might be that *S. aureus* obtained iron from hemocytes

lysed by hemolysins, which were highly upregulated at the initial infection stage in the host, as described below.

*S. aureus* has 2 manganese transporters: the *mntABC* genes encoding the ABC transporter, and the *mntH* gene encoding a proton-ion coupled transporter. Disruption mutants of both genes are reported to have reduced virulence against mice[22]. In this analysis, we found that *mntABC* genes and not the *mntH* gene were highly upregulated in the host compared with the culture medium condition (Fig. 4b). In addition, we revealed that disruption of the *mntABC* gene operon (Fig. 4c) in *S. aureus* with an intact *mntH* gene reduced virulence against mice (Fig. 4d), and the virulence was partially recovered by complementation of this operon (Supplementary Fig. 3a). These results indicated that the MntABC transporter significantly contributes to *S. aureus* virulence. In addition, a staphylopine-mediated transport system related to the acquisition of broad metal ions such as iron, zinc, copper, nickel, and cobalt was recently reported[12]. The expression level of the *cntABCDF* operon (Fig. 4e) encoding the ABC transporter increased from 24 to 48 h.p.i., and the expression level of the *cntKLM* gene, which is involved in staphylopine synthesis, increased several-fold as the infection progressed. Disruption of the *cntK* gene, which is involved in staphylopine synthesis, and the *cntE* gene, which is involved in the secretion of staphylopine from bacterial cells, significantly reduces virulence (Fig. 4f, g). Furthermore, the virulence of *cntE* and *cntK* disruptant mutants were partially and almost completely recovered in complemented strains, suggesting that this metal transporter is essential for the virulence of *S. aureus* (Supplementary Fig. 3b, c). Further studies are needed to clarify the functions of these genes in the full

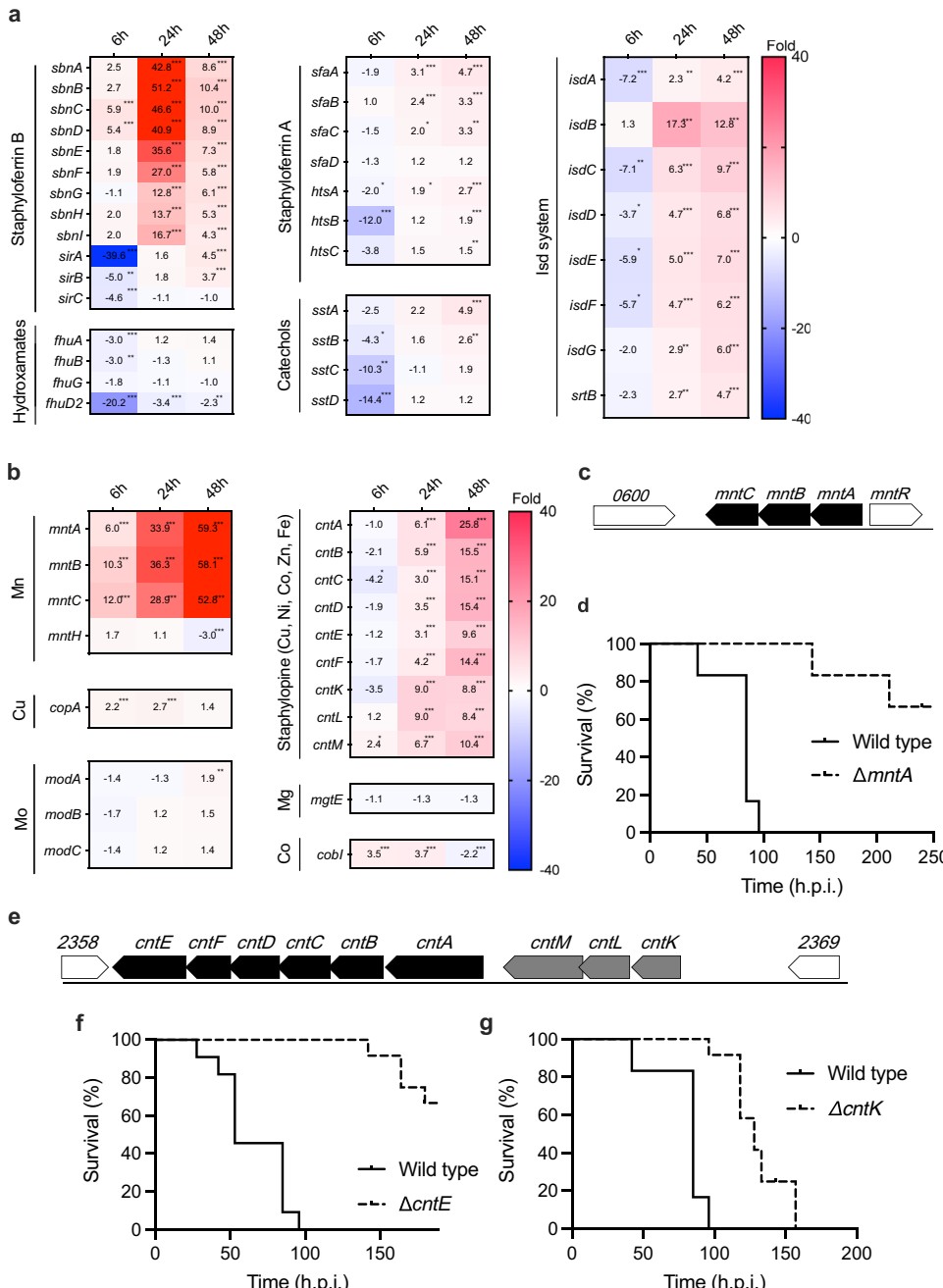

**Fig. 4 Gene expression change in _S. aureus_ metal acquisition system in the host liver compared with culture medium. a, b** Time course of expression changes in the iron acquisition systems (**a**) and the divalent cation acquisition systems (**b**) in _S. aureus_-infected liver compared with culture medium. The values in the box show fold-change compared with _S. aureus_ cultured in the TSB medium and boxes filled with red indicate >40-fold change. The asterisk *, **, and *** indicate FDR _p_-values < 0.05, <0.01, and <0.001, respectively. **c, e** Genome structure around _mntA_ gene (**c**) and _cntK_ and _cntE_ (**e**). **d, f, g** Mouse-killing ability of a disruption mutant of the _mntA_ (**d**), _cntE_ (**f**), and _cntK_ gene (**g**), respectively. Bacterial suspensions were intravenously injected. The Data were presented combined independent two experiments (_n_ = 12 for **d**, **g**, _n_ = 11 for **f**). Injection doses were 3.8 × 10⁷ for wild-type of **d**, **g**, 3.5 × 10⁷ CFU for wild-type of **f**, each 3.8 × 10⁷ CFU for Δ_mntA_, 4.0 × 10⁷ CFU for Δ_cntE_ and 3.8 × 10⁷ CFU for Δ_cntK_, respectively. Statistical analyses were performed by Log-rank (Mantel-Cox) test (**c**: _p_ = < 0.0001, chi square = 24.72, df = 1, **d**: _p_ = < 0.0001, chi square = 24.38, df = 1, **e**: _p_ = < 0.0001, chi square = 22.78, df = 1).

virulence of _S. aureus_ and to identify the responsible genes since these genes are organized into operons (Fig. 4c, e).

A gradual increase in the expression of copper and cobalt metal transporters, _copA_ and _cobI_, respectively, was observed after infection (Fig. 4b). We observed no difference in the expression of the _mgtE_ gene and _modABC_, a transporter of magnesium and molybdenum, in the host compared with that in the culture medium condition (Fig. 4b).

**Virulence factors and their regulators.** _S. aureus_ possesses a wide variety of toxins, and the expression of these toxins, such as hemolysin, increases after infection[13]. In this study, we revealed the time course of these gene expression changes (Fig. 5). Expression of genes encoding hemolysins such as _hla_ and _hlgABC_ was highly upregulated from the initial stage of infection. We assume that expression of these genes would contribute to iron acquisition by _S. aureus_ in the host at the early stage of infection.

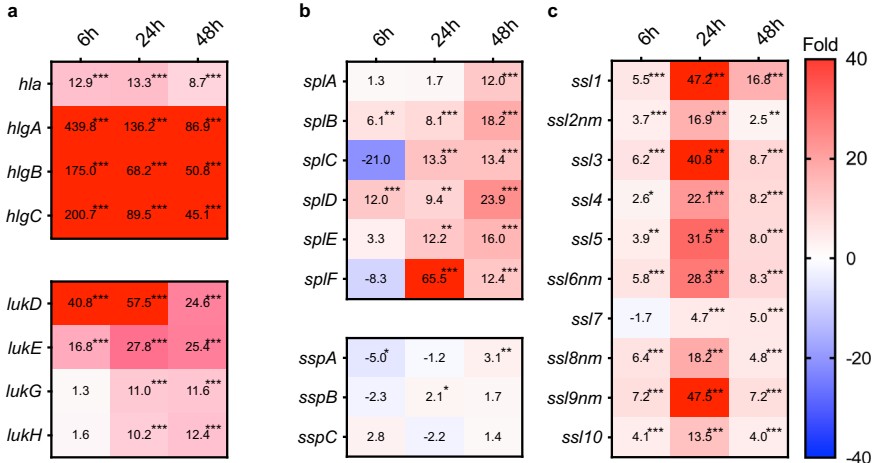

**Fig. 5 Time course of expression changes in virulence factors in *S. aureus*-infected liver compared with culture medium.** Time course of expression changes for toxins (**a**), proteases (**b**), and superantigens (**c**). Values in the boxes show fold-change compared with *S. aureus* cultured in the TSB medium and the boxes filled with red indicate >40-fold change. The asterisk *. **, and *** indicate FDR *p*-values < 0.05, <0.01, and <0.001, respectively. *lukG* and *lukH* genes in (**a**) are annotated as *lukS* and *lukF* in Refseq (NZ_CP023390.1), although we annotated these as *lukGH* genes as these genes are not related to the PVL phage and are similar to the *lukGH* gene.

The expression level of leukotoxin genes such as *lukDE* was increased from 6 h.p.i. Expression of superantigen genes, which are involved in evading the immune system[23], increased starting at 6 h.p.i., although significant expression increments were observed from 24 h.p.i. It is uncertain why the expression of genes corresponding to the host's immune response was delayed even though the innate immune system of the mice was already activated at 6 h.p.i.

We further evaluated the expression of 2-component regulatory systems identified in *S. aureus* required for environmental responses and toxin regulators. The expression of *agr* and *sae* genes, which are necessary for toxin expression[24], did not increase in the host, but rather tended to decrease at 48 h.p.i. (Table 3). These findings are consistent with findings in other models[3,25]. Several transcription factors are considered to influence *agr* gene regulation, which regulates toxin production[24,26,27]. The expression levels of *sarH1* and *sarX*, which negatively regulate *agr* gene expression, were decreased, while expression of *rsr*, a repressor of *agr* expression, was increased. On the other hand, the expression levels of *sarA*, *sarZ*, *ccpA*, and *mgrA*, which positively regulate *agr* expression, tended to decrease. Furthermore, expression of *sarH1* and *sarT*, which are negative regulators of *agr* and *sarA* genes, decreased at 6 h.p.i., but increased after 24 h.p.i. These findings suggest that the expression of transcription factors involved in the regulation of virulence gene expression in the host condition was not consistent with the interpretation of the regulation of toxin production and *agr* gene expression based on in vitro analysis. Furthermore, expression of the *tcs7RS* gene, a 2-component regulatory system with unknown function, and *kdpDE* genes, a 2-component regulatory system for potassium homeostasis, was also upregulated throughout the infection process. In addition, we recently reported a novel virulence regulator, the *yjbH* gene, which regulates the expression of an iron transporter, and several virulence factors, such as *spa* and leukotoxin, and oxidative stress response genes[28] were upregulated until 24 h.p.i., suggesting that *yjbH* contributes to the expression of virulence-related genes in the early stage of infection in mice. These results suggest that transcription of the *agr* gene was not positively regulated and other genes such as *yjbH* gene regulate the expression of virulence-related genes.

**Conclusions.** In the present study, *S. aureus* gene expression changes in the mouse liver after systemic infection were analyzed

over time by in vivo RNA-Seq. The results suggested that *S. aureus* responds to changes in oxygenation and environmental influences associated with abscess formation of *S. aureus* in the liver, thereby causing liver and heart damage. Further, only a few metal transporters were upregulated at 6 h.p.i., but many metal transport systems increased remarkably after that time. Our results also revealed the contribution of the manganese transporter *mntABC* alone and staphylopine to the pathogenicity of the broad metal transport system. As for the expression of pathogenic toxins, the timing of the upregulation differed depending on the toxin. Blood hemolytic toxin significantly increased from the early stage of infection, while the expression levels of leukocyte toxin, superantigens, and serine proteases increased in the late stage of infection. By regulating gene expression, *S. aureus* exhibits a sophisticated response to changes in environmental conditions in the host during infection.

## Methods

**Ethics statement.** The mouse experiments were performed at the University of Tokyo and Teikyo University following the regulations for animal care and use and approved by the Animal Use Committee at the Graduate School of Pharmaceutical Science at the University of Tokyo (P27-4) and Teikyo University (21-027).

***S. aureus* infection and organ collection for RNA isolation.** *S. aureus* Newman strain was grown for 20 h on TSB medium at 37 °C. The full growth was diluted 100-fold with 5 ml TSB and regrown for 24 h, and then the cells were centrifuged and suspended in PBS pH 7.2. The cells ($5.6 \times 10^7$ CFU) were injected into C57BL/6J mice via the tail vein. At 6, 24, or 48 h.p.i., mice were killed to isolate the liver. The organs were immediately placed in liquid nitrogen and maintained at −80 °C until RNA extraction. A part of the mouse organs were homogenized to calculate viable cell numbers in each organ. Each experiment was conducted with 3 animals.

**Enrichment of *S. aureus* Newman RNA from infected mouse organs and removal of Ribosomal RNA.** Mouse livers ($n = 3$ for each time-point, ~50 mg each) were homogenized and lysed in 1.5 mL of buffer RLT using an RNeasy Mini Kit (QIAGEN, Hilden, Germany) with one ϕ 5-mm zirconia bead by shaking in a 2.0-mL screw cap microtube using a Bead Crusher (μT-12, Taitec, Saitama, Japan, tube holders were chilled at −20 °C until use) at 2500 rpm for 1 min. The samples were centrifuged at 20,000 × *g* at 4 °C for 1 min, and precipitants were washed with an equal volume of PBS by centrifugation at 20,000 × *g* at 4 °C for 1 min. Bacterial cell precipitates were suspended in 200 μl of TE buffer containing 0.2 mg/mL lysostaphin and incubated at room temperature for 30 min. After adding 700 μl of RLT buffer and 200 μl volume of ϕ 0.5-mm zirconia beads, the samples were shaken at 2500 rpm for 5 min, and debris was removed by centrifugation at 5000 × *g* at room temperature for 10 min. The RNA was purified according to the manufacturer's instructions. Ribosomal RNA in the above samples was removed using a Ribo-Zero rRNA Removal Kit (Human/Mouse/Rat) (Illumina, San Diego,

**Table 3 Gene expression change in the *S. aureus* 2-component system and virulence-related transcription factors in the host liver compared with culture medium.**

| | Gene | Function | 6 h.p.i. | | 24 h.p.i. | | 48 h.p.i. | |
|---|---|---|---|---|---|---|---|---|
| | | | Fold | FDR *p*-value | Fold | FDR *p*-value | Fold | FDR *p*-value |
| Two-component factors[37] | agrC | Quorum sensing control of adhesion and virulence factors | 1 | 1 | −1.2 | 0.765 | −2.3 | 0.013 |
| | agrA | | 1.2 | 0.696 | 1.2 | 0.672 | −1.9 | 0.026 |
| | saeS | Virulence factors regulation (toxins, enzymes) | 1.1 | 0.743 | 1.1 | 0.848 | −4.7 | <0.001 |
| | saeR | | 1 | 0.996 | −1.1 | 0.918 | −4.2 | <0.001 |
| | vraR | Cell wall -affecting antibiotic resistance, cell wall biosynthesis | 2.3 | <0.001 | 1.7 | 0.024 | −2 | 0.009 |
| | vraS | | 2.1 | 0.001 | 1.3 | 0.279 | −2.6 | 0.005 |
| | graX | AMP resistance, growth at low pH | 1.9 | 0.007 | 1.1 | 0.951 | 1 | 0.996 |
| | graR | | 2.7 | <0.001 | −1 | 0.998 | −1.3 | 0.545 |
| | graS | | 1.9 | 0.007 | 1.3 | 0.464 | −1.2 | 0.694 |
| | braS | Antimicrobial peptide resistance | 1.2 | 0.588 | 1.3 | 0.365 | −1.2 | 0.581 |
| | braR | | −1.4 | 0.468 | −1.4 | 0.54 | 1.1 | 0.885 |
| | arlS | Pathogenesis mechanisms: autolysis, adhesion, biofilm | 1.4 | 0.199 | −1.2 | 0.576 | −2 | 0.016 |
| | arlR | | −1.5 | 0.18 | −2.1 | 0.022 | −1.9 | 0.023 |
| | WalR | cell wall maintenance, cell viability | −1.2 | 0.612 | −1.2 | 0.583 | −1.3 | 0.613 |
| | WalK | | 1 | 1 | −1.2 | 0.738 | −1.4 | 0.166 |
| | hptS | intercellular survival, uptake of hexose phosphate | −2.2 | 0.115 | 1.1 | 0.922 | 1.6 | 0.21 |
| | hptR | | −3.3 | 0.012 | −2.8 | 0.029 | 1.9 | 0.04 |
| | tcs7R | Uncharacterized function | 4.9 | <0.001 | 3 | 0.001 | 1.9 | 0.085 |
| | tcs7S | | 3.2 | <0.001 | 2.7 | 0.005 | 1.4 | 0.371 |
| | srrB | Anaerobic respiration, metabolism, growth at low temperature | −1.8 | 0.027 | −1.5 | 0.141 | 1.3 | 0.443 |
| | srrA | | −1.4 | 0.303 | −1.4 | 0.316 | 2.1 | 0.03 |
| | phoR | Phosphate uptake and homeostasis | 5.3 | <0.001 | 3.4 | <0.001 | 1.2 | 0.486 |
| | phoP | | 1.2 | 0.555 | −1.1 | 0.871 | 1.6 | 0.252 |
| | airR | Oxidative stress response | −1.3 | 0.634 | 1.6 | 0.272 | −1.8 | 0.089 |
| | airS | | 1.1 | 0.802 | 1 | 0.998 | −1.1 | 3831 |
| | kdpD | Potassium homeostasis regulation | 1.4 | 0.545 | 3.5 | <0.0001 | 3.7 | <0.001 |
| | kdpE | | 2.1 | 0.123 | 3.1 | 0.005 | 4.4 | <0.001 |
| | hssR | Heme metabolism regulation | −1.7 | 0.317 | −5.3 | 0.018 | 1.7 | 0.228 |
| | hssS | | 1.8 | 0.058 | −1.1 | 0.888 | −1.1 | 0.826 |
| | nreC | Response to low oxygen, nitrate reduction | 1.3 | 0.337 | 1.2 | 0.576 | 1.2 | 0.56 |
| | nreB | | 1.1 | 0.872 | 1.2 | 0.635 | 1.6 | 0.177 |
| Transcription factor[27] | sarA | Positive to *agr* expression, induction of exoproteins and repression of spa | −1.6 | 0.051 | 1.7 | 0.118 | −5.4 | <0.001 |
| | sarH1 | Repressor of *agr*, *sarA* | −13.6 | <0.001 | −8.7 | <0.001 | −4.3 | <0.001 |
| | sarU | Positive to *agr* expression, | −5.2 | 0.925 | 18.6 | 0.001 | 4.3 | <0.001 |
| | sarR | Negative to *agr*, *sarA* | 1.1 | 0.792 | 1.1 | 0.801 | −5.2 | <0.001 |
| | sarT | Negative to *agr*, *hla*, *sarU* expression | −9.4 | 0.868 | 6.9 | 0.039 | 5.3 | <0.001 |
| | sarX | Negative to *agr* | −8.6 | 0.001 | −1.2 | 0.778 | −1.7 | 0.255 |
| | sarZ | Positive to *agr*[38] | −1.7 | 0.041 | −1.5 | 0.136 | −2.6 | 0.009 |
| | CcpA | Positive to *agr* | 1.8 | 0.015 | −1.1 | 0.797 | −2.3 | 0.011 |
| | codY | Negative to *agr* | 1.8 | 0.001 | 1.4 | 0.257 | −1.1 | 0.783 |
| | mgrA | Cytoplasmic regulator; induction of efflux pumps and capsule expression; repress of surface proteins, Positive to agr, *sarZ*, *sarX* | −1.9 | 0.017 | 1.1 | 0.966 | −4.6 | <0.001 |
| | rsr | Repressor of *agr*[39] | 4.9 | <0.001 | 2.3 | 0.003 | −1.9 | 0.092 |
| | sigB | Stress response | −2.4 | <0.001 | −2.6 | <0.001 | −2.2 | <0.001 |
| | yjbH | Virulence factor regulator[28] | 5.6 | <0.001 | 2.4 | <0.001 | −1.4 | 0.139 |

CA) according to the manufacturer's protocol. For extraction of RNA from cultures in TSB medium, *S. aureus* Newman strain was grown for 20 h on TSB medium at 37 °C. The full growth was diluted 100-fold with 5 ml TSB and regrown until $A_{600} = 1.0$ (late log phase). The cells were then centrifuged and suspended in PBS pH 7.2, and library preparation was performed as described below. Four independent cultures were used for the analysis.

**Library preparation and RNA-sequencing**. RNA-Seq analysis for differential expression analysis was performed with the HiSeq platform (Illumina) or Ion Proton system (Thermo Fisher Scientific) according to the manufacturer's instructions. Briefly, for Hiseq, the RNA-Seq libraries were prepared using the TruSeq RNA sample preparation kit (version 2; Illumina), except the poly(A) selection procedure was omitted. The double-stranded PCR products were purified and size-fractionated using a bead-mediated method with AMPure XP (Beckman Coulter, California, CA). The RNA-Seq libraries were quantified by a bioanalyzer (Agilent, California, CA). Thirty-six base-pair single-end sequencing was conducted on a HiSeq 2000 or 2500 platform, using a TruSeq SR Cluster Kit v3-cBot-HS and a TruSeq SBS kit (version 3-HS). For the Ion Proton system, library preparation for RNA-Seq was performed using an Ion Total RNA-Seq Kit v2 following the manufacturer's instructions. Briefly, the ribosome depleted RNA was then fragmented by RNase III, reverse transcribed, and amplified. The size distribution and yield of the amplified library was confirmed in the bioanalyzer, and the libraries were enriched in an Ion PI Chip v2 using the Ion Chef (Thermo Fisher Scientific). Subsequent sequencing was performed in the Ion Proton System.

**Differential gene expression analysis**. All data were analyzed using CLC Genomics Workbench software, version 12 (CLC Bio, Aarhus, Denmark). Reads were aligned to the Newman genome (Accession No. NC_009641) and the mouse

genome (Mus_musculus.GRCm38) allowing a minimum length fraction of 0.95 and minimum similarity fraction of 0.95. Differential gene expression analysis was performed using edgeR analysis[29] for a normalized dataset by scaling using the TMM method[30]. Genes with an FDR $p < 0.05$ using the Benjamini and Hochberg's algorithm[31] were classified as having significantly different expression.

**Histopathology and hematological parameter analysis.** After C57BL/6J mice were infected with *S. aureus* Newman strain, livers were harvested at 6, 24, and 48 h.p.i. Histopathological analysis was performed by Advantec Co., Ltd (Osaka, Japan). Briefly, histopathological specimens were prepared by paraffin-embedding and cutting into 4-μm-thick slices. Hematoxylin and eosin staining and Gram staining (Hacker variant) were performed to evaluate the histopathology and the presence of bacterial nests in specimens. At the same time, we collected blood from the heart with a heparin-treated syringe and prepared a plasma fraction by centrifugation at $5000 \times g$ for 10 min at 4 ˚C. The samples were frozen at −80 ˚C until measurement. Hematological parameters such as ALT, AST, and CL were analyzed using a 7180 clinical analyzer (Hitachi High-Tech Corporation, Tokyo, Japan) by Ina Research Inc. (Nagano, Japan).

**Construction of *S. aureus* mutants.** Single cross-over recombination; gene disruptions were performed as previously described[32]. In summary, the internal regions within the open reading frames of the gene were amplified by PCR (Prime Star Max DNA polymerase, Takara, Tokyo, Japan) using the primers listed in Supplementary Table 3, and the PCR product was cloned into integration vector pCK20[32]. The plasmid was then transformed to *S. aureus* RN4220[26] by electroporation using Bio-Rad Genepulser Xcell (0.2 mm cubet, 2.3 kV, 100 ohm, 25 μFD). Double cross-over recombination; gene disruptions were performed as previously described[33]. Briefly, the genome DNA regions upstream and downstream of the target region were amplified by PCR using the listed primers, and then overlap extension-PCR was performed using these 2 DNA fragments together with the *aph* gene amplified from the pSF151 vector (primers: KmF; 5′-AGCGAACCATTTGAGGTGAT-3′ and KmR; 5′-GGGACCCCT ATCTAGCGAAC-3′). The PCR product was cloned into the pKOR3a vector[33] and introduced into the RN4220 strain by electroporation. Integration of the mutant cassette in the genome was confirmed by PCR and further transformed into *S. aureus* Newman[34] by phage transduction using phage 80α[35]. We confirmed that the established deletion mutants showed a similar growth curve as the wild-type (Supplementary Fig. 4). To establish the complementary strains, we amplified the *mntABC* operon, *narK* gene, *cntE* gene, and *cntLMK* operon from the genome of the Newman strain using the primers listed in Supplementary Table 3. For *narK*, *mntABC*, and *cntLMK* genes, native promoters were used and introduced into the pHY300-erm vector, in which the tetracycline-resistant gene of pHY300plk was substituted with the erythromycin-resistant gene after digestion by the restriction enzymes Hind III and BamH I. For the *cntE* gene, we introduced this gene following the *fbaA* gene promoter in pHY300-erm-fbaA, a vector that harbors the EcoRI- and BamHI-digested DNA region from the pND50-fbaA vector[36], as this gene is located at the end of the operon. The constructed plasmids were transduced to the RN4220 strain and the plasmids were transferred to each gene-disrupted mutant using bacteriophage 80α[36].

**Mouse survival assay.** *S. aureus* Newman wild-type and mutant strains were grown overnight on TSB medium supplemented with antibiotics on a rotary shaker maintained at 37 °C for 20 h to obtain full growth. The full growth was diluted 100-fold with TSB and cultured for 24 h on the same shaker, and then the cells were centrifuged and resuspended in PBS pH 7.2 to an optical density of 0.7 at 600 nm. From this, 200 μl of the cells was injected intravenously into C57BL/6J mice, and mouse survival was determined.

**Statistics and reproducibility.** Unless stated otherwise, statistical analysis was performed using GraphPad Prism version 9.0 (GraphPad Software). We used Log-rank (Mantel-Cox) test for mice survival assay and a Kruskal-Wallis test with Dunn's multiple comparisons test for the comparison of the hematological parameters. We performed more than 2 times of infection experiments as described in figure legends, except for the survival assay of complement strains in Supplementary Figs. 2a and 3.

**Reporting summary.** Further information on research design is available in the Nature Research Reporting Summary linked to this article.

## Data availability
The data were deposited in the DNA Data Bank of Japan (DDBJ) Sequence Read Archive under accession numbers DRR381801, DRR381802, DRR381803 for RNA-sequences of samples at 6 h.p.i., DRR381804, DRR381805, DRR381806 for those at 24 h.p.i. and DRR381303, DRR381304, DRR381305 for those at 48 h.p.i. Source data underlying main figures are presented in Supplementary Data 4. The other datasets generated during and/

or analyzed during the current study are available from the corresponding author on reasonable request.

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

## Acknowledgements

This work was supported by JSPS KAKENHI Grant Number JP15H05783, JP21H02733 to K.S., JSPS KAKENHI Grant Number JP24689008, JP19K07140JP, and a Grant-in-Aid for Scientific Research on Innovative Areas, Genome Science (MEXT KAKENHI Grant Number 221S0002) to H.H., in part by the Takeda Science Foundation and the Institute for Fermentation, Osaka to H.H. The illustration in Fig. 1 was utilized from DBCLS TogoTV (© 2016 DBCLS TogoTV).

## Author contributions

H.H. established and performed the in vivo RNA-Seq analysis. H.H and S.P. wrote the manuscript. S.P. and S.O. prepared the gene disruption mutants. H.H. prepared the gene complement strains. H.H., S.P., A.P., and S.O. performed the mouse systemic infection assays. Y.S. performed the RNA-Seq by Hi-Seq. K.M. gave critical aspects and comments on the histopathological analysis. K.S. critically revised the article for important intellectual content and provided final approval of the article.

## Competing interests

The authors declare competing financial interests as follows: Dr. Sekimizu is a consultant for Genome Pharmaceutical Institute Co, Ltd.
