## [Peer Review File · Communications Biology]

Reviewers' comments:

Reviewer #1 (Remarks to the Author):

Summary

The authors of this paper utilized a 2-step crush method to generate RNA sequencing data from mice infected with *Staphylococcus aureus*. Their analysis showed that many of the metabolic and virulence genes were differentially expressed at different time points during the infection. Some of the differentially expressed genes were knocked-out in subsequent experiments to verify their importance during mice infection. Overall, among other things, the authors found that the glycolytic pathway was suppressed during infection, while iron acquisition genes and nitrogen metabolism genes were upregulated. Additionally, virulence gene expression varied between different time points suggesting different strategies during different phases of mice infection.

Overall Review

This is an excellent and straightforward paper demonstrating the successful use of the 2-step crush method to get reliable in vivo RNA-seq results from *Staphylococcus aureus* infections. The paper is strengthened by the fact that the authors followed up their differential expression analysis by making mutants and re-infecting mice with the mutants to demonstrate the importance of different genes during mice infection. The data and the protocol presented herein will likely play an important role in future RNA-seq analysis of *S. aureus*. Our only major comment is about how the data is displayed in the manuscript.

Major Comments

Figures 2b, 3a-b, 4: TPM values are normalized values and in this case it is an average of multiple replicates. It is impossible to see the variance in the data or assess any statistical information. The information will be conveyed better if the authors reported the FoldChange calculated by EdgeR instead and color the boxes based on significance. Alternatively, color the values based on fold change and put '*' next to the ones that are significantly differentially expressed.

Minor Comments

Results section 1: Briefly describe the number of differentially expressed genes at each of the time points.

Line 362-363: "Differential gene expression analysis was performed using edgeR analysis for a normalized dataset by scaling using the default setting."

It is unclear what the data was normalized to. If the normalization was done with `calcNormfactor()` function in edgeR for librarysize control, it should be more clear about that.

Reviewer #2 (Remarks to the Author):

The article entitled, "Transcriptome change in *Staphylococcus aureus* in infecting mice" by Hamamoto et al utilizes newly defined methodology to shed light on the in vivo transcriptional response of *S. aureus* in the host liver during active infection. The authors examine relevant staphylococcal response signatures in pathways that control energy metabolism, virulence, and nutrient acquisition at discrete time points within the first 48 hours post-infection. These studies are coupled with a more limited evaluation of the host response signature which focuses on a subset of genes that differ more than 5-

fold in expression following infection. The study nicely highlights the adaptive nature of the *S. aureus* response to the host microenvironment in the context of a single infected organ, and lays a foundation for extension of these in vivo studies toward an investigation of other infection states and distinct tissue responses. As currently presented, the study provides only a suggestion of underlying disease mechanisms that elicit bacterial adaptation thus primarily presents an organ-specific compendium of transcriptional responses. A more substantial impact of the findings in the field may be achieved by rigorous analysis of bacterial mutant phenotypes and the host physiologic state that would enhance the conclusions that can be drawn. Several suggestions to enhance the conclusions are noted below.

Major points:

1. To draw conclusions from mortality data following infection with newly generated *S. aureus* genetic variants, a more detailed characterization of each strain is necessary including analysis of growth curves compared to the wild-type strain and genetic complementation of each mutant strain. Together, these are essential to ensure that the phenotypes observed reflect the biology of the loci of interest, and not the commonly occurring perturbation of bacterial fitness of strains that are isolated following genetic manipulation procedures.
2. Throughout the paper, there are assumptions generated about host physiology based on the transcriptional findings of *S. aureus*. For example in discussion of the *narK* mutant phenotype: "These results suggest that nitrate respiration was upregulated at the late stage of infection and required for full virulence of *S. aureus* under reduced oxygen pressure caused by progression of the infection." See also lines 295-297 "deterioration of the host's cardiovascular status". As none of these physiologic variables are rigorously assessed using clinical parameters in the study, it is important that the conclusions convey the limitation in interpretation of the findings. Alternatively, the authors may elect to more carefully dissect the host's physiologic state.
3. Related to the above, assessment of the bacterial cfu in the bloodstream and liver and liver histopathology would also be relevant biological endpoints to examine, as the *S. aureus* transcriptional response in the liver may not be mechanistically linked to the induction of mortality.

Minor points:

1. Please detail the number of complete biological replicates performed for each experiment in the figure legend and/or the methods section.
2. A review with editing to optimize grammar would be beneficial.

Reviewer #3 (Remarks to the Author):

The authors conducted RNA-seq analysis of Newman *S. aureus* in systemically infected mice, and cataloged the genes they observed to be up and down regulated as compared to *S. aureus* grown in vitro. Understanding how gene expression changes over time is important for understanding in vivo models of pathogenesis. They chose three different time points to collect livers and perform RNA-seq: 6 hours post infection, 24 hours post infection, and 48 hours post infection, and performed KEGG pathway enrichment analysis. They found significant up or down regulation in genes involved in energy metabolism, lipid metabolism, metal acquisition, virulence factors, and regulators of virulence. They determined that *narK*, involved in nitrate respiration, was important for virulence in a mouse model. Additionally, disruption of *mntA*, a gene in the *mntABC* operon that encodes the ABC transporter, also reduces lethality in mouse infections as compared to WT. Disruption of both *cntK*, a gene involved in staphylopine synthesis, and *cntE*, a gene involved in staphylopine secretion, also reduced lethality in a mouse. They show that while toxins are upregulated at various time points, *agr* and *sae* are not upregulated in vivo as compared to in vitro, and suggest that other genes are contributing to regulation of virulence factors in vivo. Unfortunately, at the end, the authors did not discover anything really new, which make the manuscript highly descriptive and more a resource paper. It is because of this that I suggest the manuscript is published in a more specialized Journal.

Major Concerns:

1. Sometimes sentences are confusing and the message the authors are trying to get across is not easily understood or even misstated. Many claims are not supported by the provided data.

i. Line 39-42 – “The number of recent discoveries of therapeutically active...”

ii. Line 205-208 – “*S. aureus* has at least 5 iron acquisition systems, and the genes involved are known to be upregulated in the host...” – this needs to be cited

iii. Small errors throughout: for example, line 24 “in vivo” is not italicized, suggesting that hla, hlgAB and hlgCB are not leukotoxins (line 254) etc.

Misleading/confusing language examples:

i. Line 216-220 and 298-300 – authors cite a paper that has shown the mnt virulence phenotype using “disruption mutants of both genes” and then say that they “revealed the contribution of the manganese transporter...for the first time.” – This was confusing until I read the cited paper and realized that they did not show a phenotype for either single mutant but saw one in a double mutant. Authors need to emphasize the double vs. single mutant different in your studies.

ii. Line 252-254 – “Expression of these genes contribute to iron acquisition of *S. aureus* in the host at the early stage of infection” – this was not shown in the paper, it is just a hypothesis or needs to be cited.

iii. Line 294-297 – “...associated with deterioration of the host’s cardiovascular status as the infection progresses.” – Where is the evidence for this? This claim is not substantiated in the paper.

iv. Line 59: “We reveal the in vivo transcriptome of *S. aureus*.” This statement is too broad, as the transcriptome that has been determined is for one infection type and one strain of *S. aureus*

v. Line 298: The authors state that “expression of metal transporters did not increase until 6 h.p.i.” however 6 hours post infection was the first timepoint that they looked at, so they cannot make this statement

2. I think it would be useful to include insight as to why they chose three timepoints and why they specifically picked 6 hours post infection, 24 hours post infection, and 48 hours post infection, as the addition of these timepoints is what makes this paper different from other in vivo IV infection RNA-seq papers that have been published in the past.

3. There are no details as to how long the authors subcultured their bacteria for before infecting mice, and there are no details as to how the authors grew their bacteria for their in vitro cultures that they then used to compare to the in vivo bacteria. This is necessary information that should be included in the manuscript.

4. Figure 4: they’re looking at lukSF expression, but Newman does not have the PVL phage and shouldn’t express lukSF.

Minor Concerns:

1. I would want to see the mouse experiment in Figure 2C repeated; there are only 5 mice in each group, and the differences between the two strains may not reproduce when repeated another time.

2. The title is awkwardly worded and a bit vague

3. Supplementary Figure 2 legend

a. Please change the colours for 24h or 48h to make them more distinct. It is not obviously clear in the figure which triangles correspond to which of these time points – especially the >2 red colouring.

Reviewer #1 (Remarks to the Author):

Summary

The authors of this paper utilized a 2-step crush method to generate RNA sequencing data from mice infected with *Staphylococcus aureus*. Their analysis showed that many of the metabolic and virulence genes were differentially expressed at different time points during the infection. Some of the differentially expressed genes were knocked-out in subsequent experiments to verify their importance during mice infection. Overall, among other things, the authors found that the glycolytic pathway was suppressed during infection, while iron acquisition genes and nitrogen metabolism genes were upregulated. Additionally, virulence gene expression varied between different time points suggesting different strategies during different phases of mice infection.

Overall Review

This is an excellent and straightforward paper demonstrating the successful use of the 2-step crush method to get reliable in vivo RNA-seq results from *Staphylococcus aureus* infections. The paper is strengthened by the fact that the authors followed up their differential expression analysis by making mutants and re-infecting mice with the mutants to demonstrate the importance of different genes during mice infection. The data and the protocol presented herein will likely play an important role in future RNA-seq analysis of *S. aureus*. Our only major comment is about how the data is displayed in the manuscript.

Thank you for your positive comments. We revised the manuscript according to your comments.

Major Comments

Figures 2b, 3a-b, 4: TPM values are normalized values and in this case it is an average of multiple replicates. It is impossible to see the variance in the data or assess any statistical information. The information will be conveyed better if the authors reported the FoldChange calculated by EdgeR instead and color the boxes based on significance. Alternatively, color the values based on fold change and put ‘*’ next to the ones that are significantly differentially expressed.

We modified these figures according to your advice. We now display these data as fold-change with statistical analysis. We also modified related sentences based on fold-change values (Figure 3,4,5). In doing so, we found a mistake in our interpretation of the data regarding the *lukDE* gene and modified related sentences (line 280-218 and Figure 5).

Minor Comments

Results section 1: Briefly describe the number of differentially expressed genes at each of the time points.

Thank you for your advice. We now describe the number of differentially expressed genes in lines 96 to 98.

Line 362-363: “Differential gene expression analysis was performed using edgeR analysis for a normalized dataset by scaling using the default setting.”

It is unclear what the data was normalized to. If the normalization was done with `calcNormfactor()` function in edgeR for librarysize control, it should be more clear about that.

We performed the normalization using the TMM method with CLC genomics workbench ver. 12 and describe it in the Methods section of the revised manuscript (line 400).

Reviewer #2 (Remarks to the Author):

The article entitled, “Transcriptome change in *Staphylococcus aureus* in infecting mice” by Hamamoto et al utilizes newly defined methodology to shed light on the in vivo transcriptional response of *S. aureus* in the host liver during active infection. The authors examine relevant staphylococcal response signatures in pathways that control energy metabolism, virulence, and nutrient acquisition at discrete time points within the first 48 hours post-infection. These studies are coupled with a more limited evaluation of the host response signature which focuses on a subset of genes that differ more than 5-fold in expression following infection. The study nicely highlights the adaptive nature of the *S. aureus* response to the host microenvironment in the context of a single infected organ, and lays a foundation for extension of these in vivo studies toward an investigation of other infection states and distinct tissue responses. As currently presented, the study provides only a suggestion of underlying disease mechanisms that elicit bacterial adaptation thus primarily presents an organ-specific compendium of transcriptional responses. A more substantial impact of the findings in the field may be achieved by rigorous analysis of bacterial mutant phenotypes and the host physiologic state that would enhance the conclusions that can be drawn. Several suggestions to enhance the conclusions are noted below.

Thank you for your critical comments. We performed several additional experiments and incorporated the data into the revised manuscript. These comments helped us to significantly improve our manuscript and extend our hypothesis.

Major points:

1. To draw conclusions from mortality data following infection with newly generated *S. aureus* genetic variants, a more detailed characterization of each strain is necessary including analysis of growth curves compared to the wild-type strain and genetic complementation of each mutant strain. Together, these are essential to ensure that the phenotypes observed reflect the biology of the loci of interest, and not the commonly occurring perturbation of bacterial fitness of strains that are isolated following genetic manipulation procedures.

Thank you very much for your comments. We agree that more detailed characterization of each candidate gene identified is necessary. As suggested, we compared the growth of each strain to test if the fitness was affected by gene manipulation (Supplementary Figure 3). We found that all gene disruptant mutants exhibited a similar growth pattern as the wild-type, suggesting that the gene disruption did not affect normal growth (line 441-442). We are now working on a detailed characterization and mechanistic analysis of each identified virulence factor, including complementation of each mutant strain. Nevertheless, there are limitations regarding gene responsibility for virulence, which we note in lines 184-185, 251-253.

2. Throughout the paper, there are assumptions generated about host physiology based on the transcriptional findings of *S. aureus*. For example in discussion of the *narK* mutant phenotype: “These results suggest that nitrate respiration was upregulated at the late stage of infection and required for full virulence of *S. aureus* under reduced oxygen pressure caused by progression of the infection.” See also lines 295-297 “deterioration of the host’s cardiovascular status”. As none of these physiologic variables are rigorously assessed using clinical parameters in the study, it is important that the conclusions convey the limitation in interpretation of the findings. Alternatively, the authors may elect to more carefully dissect the host’s physiologic state.

3. Related to the above, assessment of the bacterial cfu in the bloodstream and liver and liver histopathology would also be relevant biological endpoints to examine, as the *S. aureus* transcriptional response in the liver may not be mechanistically linked to the induction of mortality.

According to your suggestion, we evaluated hematological parameters such as ALT, AST and CL. These values were significantly increased at 24 and 48 h.p.i, and not at 6 h.p.i., suggesting that the liver and heart were damaged by 24 h.p.i. These results well correlated with the respiratory status of

S. aureus. In addition, we assessed the liver histopathology. A number of abscesses were observed at 24 and 48 h.p.i., but not at 6 h.p.i. The number of bacteria in the abscesses increased in a time-dependent manner. These results suggested that *S. aureus* colonized in the liver by 24 h.p.i. and caused liver damage, which may lead to an insufficient supply of oxygen. We now discuss these points in line 145-153 and 179-184.

We further evaluated the bacterial burden in the blood, but only a very small number of bacteria were observed throughout the period. We added the data in Figure 2b (previous Supplementary Figure 1b) and a description in line 89-90.

Minor points:

1. Please detail the number of complete biological replicates performed for each experiment in the figure legend and/or the methods section.

We added the number of biological replicates to the legend for Figure 2b-f, 3c, 4d, f, g and Supplementary Figure 2. For RNA-Seq analysis, we added number of samples used in the Material and Method section (line 357 and 371-372).

2. A review with editing to optimize grammar would be beneficial.

The revised manuscript was edited by professional native-English speaking science editors.

Reviewer #3 (Remarks to the Author):

The authors conducted RNA-seq analysis of Newman *S. aureus* in systemically infected mice, and cataloged the genes they observed to be up and down regulated as compared to *S. aureus* grown in vitro. Understanding how gene expression changes over time is important for understanding in vivo models of pathogenesis. They chose three different time points to collect livers and perform RNA-seq: 6 hours post infection, 24 hours post infection, and 48 hours post infection, and performed KEGG pathway enrichment analysis. They found significant up or down regulation in genes involved in energy metabolism, lipid metabolism, metal acquisition, virulence factors, and regulators of virulence.

They determined that narK, involved in nitrate respiration, was important for virulence in a mouse model. Additionally, disruption of mntA, a gene in the mntABC operon that encodes the ABC transporter, also reduces lethality in mouse infections as compared to WT. Disruption of both cntK, a gene involved in staphylopine synthesis, and cntE, a gene involved in staphylopine secretion, also reduced lethality in a mouse. They show that while toxins are upregulated at various time points, agr and sae are not upregulated in vivo as compared to in vitro, and suggest that other genes are contributing to regulation of virulence factors in vivo. Unfortunately, at the end, the authors did not discover anything really new, which make the manuscript highly descriptive and more a resource paper. It is because of this that I suggest the manuscript is published in a more specialized Journal.

Thank you for your critical comments. We revised the manuscript according to your comments.

Major Concerns:

1. Sometimes sentences are confusing and the message the authors are trying to get across is not easily understood or even misstated. Many claims are not supported by the provided data.

i. Line 39-42 – “The number of recent discoveries of therapeutically active...”

Thank you for your kind advice. We revised this sentence for clarity and cite the references.

ii. Line 205-208 – “S. aureus has at least 5 iron acquisition systems, and the genes involved are known to be upregulated in the host...” – this needs to be cited

According to your suggestion, we cited the appropriate references (now line 227-228).

iii. Small errors throughout: for example, line 24 "in vivo" is not italicized, suggesting that hla, hlgAB and hlgCB are not leukotoxins (line 254) etc.

We checked throughout the manuscript and corrected these errors.

Misleading/confusing language examples:

i. Line 216-220 and 298-300 – authors cite a paper that has shown the mnt virulence phenotype using “disruption mutants of both genes” and then say that they “revealed the contribution of the manganese transporter...for the first time.” – This was confusing until I read the cited paper and realized that they

did not show a phenotype for either single mutant but saw one in a double mutant. Authors need to emphasize the double vs. single mutant difference in your studies.

According to your suggestion, we rewrote these sentences to improve the clarity (now line 237-243 and line 330).

ii. Line 252-254 – “Expression of these genes contribute to iron acquisition of *S. aureus* in the host at the early stage of infection” – this was not shown in the paper, it is just a hypothesis or needs to be cited.

It is our hypothesis, and this is now clarified (now line 278-280)

iii. Line 294-297 – “...associated with deterioration of the host’s cardiovascular status as the infection progresses.” – Where is the evidence for this? This claim is not substantiated in the paper.

We performed histopathology of liver and evaluated the hematological parameters of the liver and heart. A number of abscesses were observed at 24 and 48 h.p.i., but not at 6 h.p.i. The number of bacteria in the abscess increased in a time-dependent manner. In addition, hematological parameters such as ALT, AST, and CK, indicating liver and heart damage, were highly increased at 24 and 48 h.p.i. These results suggested that *S. aureus* had colonized in the liver by 24 h.p.i. and caused liver and heart damage, which may lead to an insufficient supply of oxygen. We modified this sentence according to these findings (now line 325-327).

iv. Line 59: "We reveal the in vivo transcriptome of *S. aureus*." This statement is too broad, as the transcriptome that has been determined is for one infection type and one strain of *S. aureus*

We now specify the infection type, observed organ, and strain (now line 58-68).

v. Line 298: The authors state that "expression of metal transporters did not increase until 6 h.p.i." however 6 hours post infection was the first timepoint that they looked at, so they cannot make this statement

We agree with this point and modified this sentence to be consistent with the results (now line 328-329).

2. I think it would be useful to include insight as to why they chose three timepoints and why they specifically picked 6 hours post infection, 24 hours post infection, and 48 hours post infection, as the addition of these timepoints is what makes this paper different from other in vivo IV infection RNA-seq papers that have been published in the past.

According to this suggestion, we added an explanation for why we selected these time-points and the differences from other in vivo RNA-Seq analyses (line 65-68).

3. There are no details as to how long the authors subcultured their bacteria for before infecting mice, and there are no details as to how the authors grew their bacteria for their in vitro cultures that they then used to compare to the in vivo bacteria. This is necessary information that should be included in the manuscript.

According to your suggestions, we now include detailed culture conditions for the infection assay and control RNA for the in vitro culture. Thank you for pointing out the lack of important information.

4. Figure 4: they're looking at lukSF expression, but Newman does not have the PVL phage and shouldn't express lukSF.

Thank you for your comments. In fact, Newman *lukSF* genes are not related to the PVL phage, but these genes are annotated as *lukSF* (leukocidin/hemolysin toxin subunit S/F) in the RefSeq file. From the Blastx results, the protein sequences were suggested to be *lukGH*, and thus we changed the name of the gene in this manuscript and made a note regarding the name in the legend for Figure 5.

Minor Concerns:

1. I would want to see the mouse experiment in Figure 2C repeated; there are only 5 mice in each group, and the differences between the two strains may not reproduce when repeated another time.

We repeated the $\Delta narK$ strain infection assay and obtained the same result.

2. The title is awkwardly worded and a bit vague

According to your suggestion, we changed the title.

3. Supplementary Figure 2 legend a. Please change the colours for 24h or 48h to make them more distinct. It is not obviously clear in the figure which triangles correspond to which of these time points – especially the >2 red colouring.

We changed the colors to be more distinct, according to your suggestion.

Reviewers' comments:

Reviewer #2 (Remarks to the Author):

The authors have largely addressed the points raised in the initial review, and this has improved the manuscript. As noted previously, the study is of interest in the field, and illustrates both new technologic and scientific insight on the host-pathogen interaction. For data on newly generated bacterial mutant strains to be presented and rigorously evaluated, however, it is imperative to generate and evaluate genetically complemented strains to demonstrate that the virulence properties ascribed to the mutation are indeed manifestations of the single gene deletion event. This is of keen importance for genes such as those examined in the study wherein disrupted loci control metabolism or exist in broader operons (highly relevant to the *mnt* and *cnt* loci). While growth in complete medium (as demonstrated in the new supp figure) is essential to examine new strains for significant growth aberrations, this does not supplant the need for genetic complementation for in vivo studies. The extension of these studies to include these essential controls and validation of the data/conclusions would render this study suitable for publication in Nat Comm.

Reviewer #3 (Remarks to the Author):

Overall, the revised manuscript is much improved. However, there are still several issues that must be addressed.

1. The lack of complementation studies is a huge issue. *S. aureus*, including strain Newman, is notorious for acquiring spurious mutations during the in vitro mutagenesis process. As such, complementation studies are required to establish the critical role of *nark*, *mntA*, *cntE* and *cntK*. Complementation studies are possible by replacing the KO locus with the WT gene. Alternatively, the authors could sequence their mutants (100 x coverage) and show that there are no unintended mutations.

2. In vivo data (Fig 3c, 4d, 4f, and 4g): while I appreciated that the authors now say the data are representative of independent experiments, the appropriate way to show survival data from multiple experiment is to combine the data and show a single graph with all the data ($n = >8$ mice from at least 2 independent experiments). This is important as otherwise the reader don't have a way to evaluate the reproducibility of the findings.

3. Methods: The section of ribosomal RNA extraction needs additional details to enable others to use the protocol. For instance, how much RLT and 5-mm beads were used to homogenize the livers? Was the entire liver used? In what kind of tubes was this done? The speed of the centrifugations (g force) is also lacking.

Minor:

Page 12, line 196> Replace "treated" with infected.

Rebuttal letter

We thank the editor and reviewers for their critical comments on the complementation of disrupted genes. We established complementary strains and performed an infection assay. We found that the virulence of each mutant was recovered, except for that of the *narK* gene-disrupted mutant. We therefore could not observe the recovery of virulence in a complementary strain (see the following figure). Therefore, we deleted the results regarding the $\Delta narK$ strain from the manuscript. We respectfully acknowledge both reviewers' appropriate comments.

Figure| Survival curves of mice injected with wild-type, *narK* gene-disrupted strains, and the $\Delta narK/narK$ complement strain via tail vein. For the wild-type and $\Delta narK$ strains, the results of 3 independent experiments were combined (wild-type: 6.0×10^7 , 4.3×10^7 and 3.7×10^7 n=17, $\Delta narK$: 6.0×10^7 , 4.6×10^7 and 3.9×10^7 n=16, respectively) and for the $\Delta narK/pnarK$ strain, only a single experiment was performed (4.8×10^7 CFU n=6). Statistical analysis was performed by the log-rank test ($p=0.0003$ chi square=13.72, df=1 between WT and $\Delta narK$ strain, and $p=0.2635$ chi square=1.250, df=1 between the $\Delta narK/pnarK$ and $\Delta narK$ strains).

Reviewers' comments:

Reviewer #2 (Remarks to the Author):

The authors have largely addressed the points raised in the initial review, and this has improved the

manuscript. As noted previously, the study is of interest in the field, and illustrates both new technologic and scientific insight on the host-pathogen interaction. For data on newly generated bacterial mutant strains to be presented and rigorously evaluated, however, it is imperative to generate and evaluate genetically complemented strains to demonstrate that the virulence properties ascribed to the mutation are indeed manifestations of the single gene deletion event. This is of keen importance for genes such as those examined in the study wherein disrupted loci control metabolism or exist in broader operons (highly relevant to the *mnt* and *cnt* loci). While growth in complete medium (as demonstrated in the new supp figure) is essential to examine new strains for significant growth aberrations, this does not supplant the need for genetic complementation for *in vivo* studies. The extension of these studies to include these essential controls and validation of the data/conclusions would render this study suitable for publication in Nat Comm.

Thank you for your critical comments. We established a complement strain and performed survival analysis using mice according to your advice. The data were added to Supplementary Figure 3.

Reviewer #3 (Remarks to the Author):

Overall, the revised manuscript is much improved. However, there are still several issues that must be addressed.

1. The lack of complementation studies is a huge issue. *S. aureus*, including strain Newman, is notorious for acquiring spurious mutations during the *in vitro* mutagenesis process. As such, complementation studies are required to establish the critical role of *nark*, *mntA*, *cntE* and *cntK*. Complementation studies are possible by replacing the KO locus with the WT gene. Alternatively, the authors could sequence their mutants (100 x coverage) and show that there are no unintended mutations.

Thank you for your advice. As mentioned above, we established a complement strain and performed a survival assay. The results were incorporated in Supplementary Figure 3.

2. *In vivo* data (Fig 3c, 4d, 4f, and 4g): while I appreciated that the authors now say the data are representative of independent experiments, the appropriate way to show survival data from multiple experiment is to combine the data and show a single graph with all the data ($n = >8$ mice from at least 2 independent experiments). This is important as otherwise the reader don't have a way to evaluate the

reproducibility of the findings.

Thank you for your valuable comments. We combined the results and provided a single figure in Supplementary Figure 2 and Figure 4d, f, g.

3. Methods: The section of ribosomal RNA extraction needs additional details to enable others to use the protocol. For instance, how much RLT and 5-mm beads were used to homogenize the livers? Was the entire liver used? In what kind of tubes was this done? The speed of the centrifugations (g force) is also lacking.

Thank you for your suggestion. We added the information regarding RNA isolation from the organs (lines 367-377).

Minor:

Page 12, line 196> Replace "treated" with infected.

Thank you. The data referred to in this sentence were removed from the manuscript.

REVIEWERS' COMMENTS:

Reviewer #3 (Remarks to the Author):

The revised manuscript is now suitable for publication.